# Flexible and Robust Triboelectric Nanogenerators with Chemically Prepared Metal Electrodes and a Plastic Contact Interface Based on Low-Cost Pressure-Sensitive Adhesive

**DOI:** 10.3390/s23042021

**Published:** 2023-02-10

**Authors:** Shuai-Chen Wang, Binbin Zhang, Lijing Kang, Cunman Liang, Dongdong Chen, Guoqiang Liu, Xuyun Guo

**Affiliations:** 1Department of Electronic Engineering, The Chinese University of Hong Kong, New Territories, Hong Kong SAR 999077, China; 2Hong Kong Center for Cerebro-Cardiovascular Health Engineering (COCHE), Shatin, New Territories, Hong Kong SAR 999077, China; 3Biomedical Engineering, The City University of Hong Kong, New Territories, Hong Kong SAR 999077, China; 4Epro Advance Technology Limited, Hong Kong Factory, 35 Wang Lok Street, Yuen Long Industrial Estate, New Territories, Hong Kong SAR 999077, China; 5Center of Advanced Lubrication and Seal Materials, State Key Laboratory of Solidification Processing, Northwestern Polytechnical University, Xi’an 710072, China; 6School of Chemistry, Centre for Research on Adaptive Nanostructures and Nanodevices (CRANN) & Advanced Materials Bio-Engineering Research Centre (AMBER), D02 PN40 Dublin, Ireland

**Keywords:** wearable devices, triboelectric nanogenerator, electroless deposition, polymer-assisted metal deposition, real contact area

## Abstract

Triboelectric nanogenerators (TENGs) are devices that can harvest energy from mechanical motions; such devices can be used to power wearable sensors and various low-power electronics. To increase the lifetime of the device, scientists mainly use the method of making TENG in a hard skeleton to simplify the complex possible relative movements between two triboelectric parts. However, the hard skeletons cannot be embedded in soft and lightweight clothing. To make matters worse, the materials used in the garments must be able to withstand high mechanical forces when worn, such as the pressure of more than 100 KPa exerted by body pressure or everyday knocks. Notably, the TENGs are usually made of fragile materials, such as vacuum-evaporated metal electrodes and nano-sized coatings, on the contact interface; these electrodes and coatings often chip or wear off under the action of external loads. In this work, we succeeded in creating a thin, light-weight, but extremely robust garment-integrated triboelectric nanogenerator (G-TENG) that can be embedded in clothing and pass the water wash test. First, we chemically deposited a durable electrode with flexible properties for G-TENG using a novel technique called polymer-assisted metal deposition (PAMD). The as-formed metal electrodes are firmly bonded to the plastic substrate by a sub-10 nm adhesive polymer brush and can withstand a pressure of 22.5 MPa and a tear force of 0.7 MPa. We then removed the traditionally used fragile nanoparticle materials and the non-durable poly-dimethylsiloxane (PDMS) layer at the triboelectric interface, and then used a cost-effective, durable and slightly flowable pressure-sensitive adhesive to form a plastic contact interface. Such a soft plastic interface can ensure full contact of the triboelectric materials, which is excellent in complex environments and ultimately improves the power generation efficiency of the devices. The as-formed low-cost energy harvesting device could become an industry standard for future smart clothing.

## 1. Introduction

Triboelectric nanogenerators (TENGs) [1,2,3,4,5,6] are a group of flexible energy-harvesting devices that have attracted considerable research attention due to their potential use as power sources [7,8] or sensing systems [9,10,11] to be embedded into smart garments [12]. The invention of this device, pioneered by Wang et al., has been developed into wearable devices that are light-weight and highly adherent to the skin [13,14], and the adhesion of the device can even be adjusted by temperature to achieve truly comfortable wear [15]. In the future, it is hypothesized that TENGs will be widely used to generate transient electrical power for wearable electronics; this widespread use will be accelerated by development of low-power wearable devices and flexible electronics [16] and integration of an electronic system, such as microelectromechanical systems (MECS) chips [17], a wireless transmission chip [7], and electrowetting on a dielectric (EWOD) chip [18].

However, wearable and flexible TENGs usually suffer from low durability and short device lifetime. The most obvious reason is that the electrode material used for TENGs is not durable because the metal electrodes of TENGs are usually deposited by vacuum evaporation techniques, which lack an adhesive layer and are easily delaminated from the plastic substrates when the whole device undergoes large deformation [19].

Another reason may be the fragility of the nanomaterials and deformable materials used in the triboelectric interface region of TENGs.

Addition of plastic and soft material is necessary for TENGs because the surfaces of normal objects are rough, which prevents full contact between two triboelectric layers; the presence of many raised structures can make it difficult for a contact interface to achieve full contact without the assistance of the deformable features of plastic materials. In addition, airborne particles or dust (e.g., PM 2.5) can be deposited on the surfaces of two triboelectric materials and prevent full contact.

Therefore, many scientists choose to use a soft triboelectric layer, e.g., polydimethylsiloxane (PDMS), to fabricate a triboelectric interface [20]. The PDMS material is often used in conjunction with the nanomaterials that are embedded or attached to another triboelectric surface of the TENG [21]; the surface of the soft PDMS will meet the surface of the opposite hard material to deform and fit the opposite nanostructures, thereby increasing the real contact area of the TENG, i.e., increasing the surface area of the contact interface and making it larger than the ideal flat contact area.

However, the key situation is that both nanoparticles and PDMS thin film are fragile; the nanoparticles are easily worn out after repeated triboactivations, while the soft PDMS thin film is prone to delamination or detachment when the device undergoes large deformations. The properties of the triboelectric interface lead to damage, breakage, or detachment of the soft material (i.e., the PDMS) when acted upon by the large mechanical forces that the clothing material must bear, such as torques larger than 8 kg·m^2^/s^2^ during machine washing or pressures of >100 KPa that can occur when the garment is worn (e.g., an adult male weighing 75 kg uses his elbows to support his entire body weight with only ~ 36 cm^2^ of contact area between the elbow and the flat floor).

We know that the intrinsic surface energy of PDMS is very low [22], it is difficult to adhere to the surface of ordinary objects, and it will chip off under large external pressure and scratches. In addition, the intrinsic mechanical strength of PDMS is also not too high, so the PDMS is also easily broken from the inside of the structure or cracked under external stress. To increase the lifetime of the device, scientists have invented various hard skeletons [23,24] to protect the materials at the triboelectric interface. The hard skeletons limit the relative motion of two triboelectric parts from unpredictable relative motion to simple relative motion (e.g., in contact separation mode [25] or sliding mode [26]), with the aim of preventing the fragile PDMS material and nanomaterials from delaminating or detaching when the device is subjected to large external forces. However, the hard skeletons cannot be embedded in flexible clothing materials as this type of hard substrate does not complement the design concept of soft and lightweight electronic clothing.

Therefore, replacing the existing materials is the only practical option to increase the life of the device. In this article, we propose the use of different materials to increase the life of a flexible G-TENG.

In manufacturing the flexible and durable G-TENG, we started by replacing the original vacuum-coated electrodes with new highly adhesive and durable electrodes prepared by the chemical solution method, namely polymer-assisted metal deposition (PAMD) techniques [27,28,29,30,31]. PAMD technology is attracting increasing attention from the business and research communities as an emerging technology for the production of highly adherent metal thin films on plastic substrates. PAMD-based thin-film electrodes, which are characterized by a polymer adhesion layer on the underside, have good adhesion to the surface of a plastic substrate; it is inherently difficult for other thin-film coatings to form an adhesion force. We have also observed for the first time that our thin-film electrodes have a 1–10 nm thick polymer brush adhesion layer. Due to the ultrathin nature of this adhesion layer, the electrodes are tightly bonded to the polymer surface, which differs from the mechanical behaviors of electrodes that are normally held together with thick adhesive materials.

In addition, this reported solution method for the preparation of metal thin-film electrodes at room temperature has many advantages over metal thin-film electrodes prepared by the now widely used method of printing metal thin-film electrodes, i.e., printed metal nanoparticle ink technology [32], because nanoparticle inks do not work well on many heat-sensitive plastic substrates, such as polyethylene terephthalate (PET) and polyethylene (PE), because they require a high-temperature (>150 °C) post-treatment step [33], which causes irreversible damage to the plastic substrates.

Although the PAMD technique has great potential in production of high-quality TENGs, the existing report is somewhat stereotypical; it only reports application of this technique in production of textile-structure-based TENGs [34,35], which has been demonstrated by previous PAMD-related works [36,37]. However, textiles and clothing are prone to pilling when worn too much. In addition, TENGs based on textile structures have too many contact points; even a small piece of textile can have hundreds of contact points. If the insulating layer of a contact point is worn out, it will cause a short circuit or failure of the whole device.

Considering the above reasons, our TENG devices turn fiber-to-fiber contact into wide face-to-face contact; the PAMD-based metal electrodes were grown on a uniform large area rather than growing electrode coatings that wrap around fibers or yarns. The advantage of this design is that the enormous external pressure is distributed over a larger contact area rather than being concentrated on a small contact area. As such, this design provides possibility for application of PAMD in the field of durable and washable TENGs.

In addition, the actual power generation mechanism of the TENG needs further research. The electrical energy of the TENG device is generated in the contact area, but, in recent years, the progress of research on the contact area of the TENG has been relatively slow. Only in 2020, the concept of the real contact area was emphasized by Xu et al. [38], and, in 2022, Wen et al. started to introduce a variable describing roughness of triboelectric material in the TENG mold to accurately estimate the equivalent capacitance that can be generated by the TENG device [39].

Soft materials are always used at the contact interface of the two triboelectric materials in order to increase the real-contact area [40]; by increasing the real-contact area, the voltage and currents generated by the TENG are increased. Sometimes, a soft material is used together with another rough interface with the inlay of rigid nanoparticles on its surface [38]. As a result of deformation of the soft interface induced by an externally applied force, the soft material can adapt to the conformation of the rough surface, thus increasing the real contact area.

Second, we redesigned the materials used to make the triboelectric interface of the TENG; we removed the fragile nanomaterials in the contact interface. We then used a low-cost but more durable pressure-sensitive adhesive to replace the brittle and easily detachable PDMS material, with the aim of creating a reliable contact interface that can adapt to complex environments and increase the output current of the device, which is another key point of our article.

The soft pressure-sensitive adhesive will automatically fill the uneven surface area of the opposite substrate during each contact; the softer pressure-sensitive adhesive will flow into the concave part of the micro-nano gap on the surface of the opposite ordinary polymer material to achieve full contact. The real friction situation can be more complicated than imagined. In everyday life, there are various impurities or dust that fall on the surface of clothing. A soft and durable pressure-sensitive adhesive with certain flow characteristics will wrap around any impurities or protrusions so that the actual contact area is close to the ideal plane of actual contact.

Working in conjunction with the chemically produced durable metal electrodes, the resulting soft and light-weight TENG was extremely durable and could be adapted to complex mechanical motion environments, e.g., working in motion in a combination of contact separation mode and sliding mode. In addition, the flexible TENG could be sewn directly into clothing to generate electrical power for many possible low-power electronics.

As a proof of concept, we carried out a detailed analysis of the mechanical and electrical properties of the chemically prepared metal electrodes and found that the as-formed copper electrode has good adhesion to the plastic substrate and can pass the pull-off test. The as-formed metal coating will not break under a maximum tear force of 0.7 MPa and pressure of 22.5 MPa. At the same time, it has excellent conductivity and low surface roughness. The sheet resistance of the electrode can reach about 1 Ω/□. The root-mean-square roughness (Rq) of the electrode was measured to be ~3 nm, and the conductivity of the electrode gradually became constant during the 36 h long-term continuous bending test, the bending radius being as small as 1 mm. In addition, a soft material, i.e., the pressure-sensitive adhesive, was used to replace the PDMS as the triboelectric interface in order to increase the real contact area of the interface. This material was found to be stable when subjected to bending or water wash test. In the water wash test, the degradation of the output performance dropped by less than 20%, and, beyond this initial degradation, the performance was found to be stable again. Application of these high-strength film materials provides a good technical model for future mechanical energy-harvesting units that can withstand high impact or twisting and be embedded on the surface of clothing.

## 2. Materials and Methods

### 2.1. Fabrication of the Chemically Prepared Metal Electrode on the Flexible Triboelectric Materials Aims to Form Super-Durable and Flexible TENG

The PE film (with its adhesive layer) and the 50 µm thick PET film were first bonded together. In this way, two triboelectric layers were bonded together to protect the contact interface from exposure to the chemical solutions during the growth of metal electrodes. This sandwich structure, two triboelectric films with a pressure-sensitive adhesive in the middle, was then dipped directly into the chemical solution for chemically deposition of the metal electrodes.

The hydrophilic surfaces of the PET and PE substrates were first subjected to an oxygen–oxygen plasma treatment for 4 min. They were then immersed in a 3-(trimethoxysilyl) propyl methacrylate (MPS) (0.5 wt%, Sigma-Aldrich, Burlington, VT, USA) alcohol solution for 1 h. During this time, an MPS-based self-assembled monolayer (SAM) formed on the substrate surface. After ethanol rain treatment and subsequent drying, the SAM-modified substrate was obtained. Subsequently, the polymer brush of 2-(methacryloyloxy) ethyl-trimethylammonium chloride (METAC) (Sigma-Aldrich, Burlington, VT, USA) was grown on the acrylate group of MPS in an aqueous solution (150 g/L, Sigma-Aldrich, Sigma-Aldrich, Burlington, VT, USA) at a concentration of 20 wt% mixed with the initiator potassium persulfate (KPS) (2.5 g/L, Sigma-Aldrich, Burlington, VT, USA). The METAC was grown at 80 °C for 1 h. The PMETAC-modified substrates were then rinsed with deionized water and dried by spraying with pure nitrogen. An aqueous solution of 5 mM ammonium tetrachloropalladate (II) was used to treat the polymer-brush-based copper electrode for 30 min. Electroless metal deposition was then carried out using a freshly prepared electroless deposition (ELD) solution. In this process, the thickness of the metal electrode can be controlled by varying the duration of the metal electrode deposition.

### 2.2. Preparation of the Soft Contact Interface Using a Low-Cost Pressure-Sensitive Adhesive

PET was chosen as the electron acceptor layer for the formed TENG, and a low-cost rubber (poly-cis-isoprene) adhesive (Hao Jing, Taobao, Shenzhen, China) was used as the electron donor layer [41,42]. First, a roll of clean polyethylene (PE) film with a thickness of about 40 µm was purchased from Taobao (Hong Shun, Guangzhou, China). Second, using the roll-to-roll method, a thin layer of hot-melt adhesive film made of low-cost rubber (poly-cis-isoprene) (Hao Jing, Taobao, Shenzhen, China) is quickly hot-pressed onto the surface of the PE film by a hot-pressing roller. The exposed adhesive can form a strong and durable contact interface when combined with a transparent release film of polyethylene terephthalate (PET) (Chang Feng, Taobao, Guangzhou, China).

## 3. Results

### 3.1. Chemical Fabrication of the Metal Electrode on Flexible Triboelectric Materials Aims to Form a Super Durable and Flexible TENG

#### 3.1.1. Simultaneously Prepare the Dual Electrodes of the TENG by Chemical Methods

Unlike the thermally evaporated metal electrode, which has no bonding material between the electrode and the plastic substrate, the chemically prepared metal electrodes have a polymer brush layer grown by Atom Transfer Radical Polymerisation (ATRP) technique under the metal electrode, which can provide a strong adhesion force for the metal electrode.

Here, we prepared a highly conductive and durable metal electrode in solution environment. In this process, two triboelectric materials were firstly first bonded together by using a pressure-sensitive adhesive, with only one side of the plastic dielectric layer was exposed to the chemical solutions, as shown in Figure 1a. Once the surface of the exposed dielectric layer has been covered by durable metal electrodes, a complete two-piece TENG device is formed. This process is suitable for industrial production of TENGs.

The oxygen plasma treatment was applied to the two exposed dielectric layers to increase the number of hydroxyl groups on the surfaces and their wettability to an aqueous solution. This step was necessary because the surfaces of the two dielectric layers had intrinsically different wettability as the PET molecule contains hydroxyl groups while the PE molecule does not (as shown in Figure 1b). The oxygen plasma treatment generates hydroxyl groups on the PE surface and also increases the density of hydroxyl groups on the PET surface, giving both sides good wettability.

Atom transfer radical polymerization (ATRP) [43] is an effective method that enables radical polymerization at a controllable rate, which permits growth of polymer brushes. Surface-initiated ATRP (SI-ATRP) [44] is a related technique that allows polymerization to take place stably in different substrates. As shown in Figure 1c, in this work, MPS with a silane group was used to create a bond between the smooth surface of the plastic materials (PE and PET); an acrylate group in the MPS enabled graft polymerization. As shown in Figure 1d, the oxidant KPS was used as the initiator for the SI-ATRP, which resulted in formation of the METAC-based polymer brush. The reaction was carried out in an aqueous solution over a period of 1 h.

The cationic polymer brush was used to immobilize the catalyst anions of ammonium tetrachloropalladate (II), as shown in Figure 1e. The PbCl^2−^ containing palladium metal catalyst anions gradually replaced the original chlorine ion paired with the cationic polymer brush during a long-term immersion process. Ion replacement was achieved in a 1 mM aqueous solution of ammonium tetrachloropalladate (II) over a period of more than 20 min.

Finally, the immobilized palladium metal ions were used as a catalyst to deposit copper on the polymer brush; in this way, the copper formed a strong and reliable thin-film electrode on two dielectric layers and built up all the necessary components of the TENG device (Figure 1f).

#### 3.1.2. Tearing of the Triboelectric Structure to Form the TENG

As shown in Figure 2a, a layer of copper was deposited on the surface of the PE and PET films. The two films, bonded together with a pressure-sensitive adhesive, separated along the adhesive/PET interface to form two triboelectric parts of the TENG, as shown in Figure 2b. Using a conductive copper tape to connect wires to the electrodes, the combination and separation of these two films generated sufficient electrical energy to power a light-emitting diode (LED) lamp, as shown in Figure 2c–e.

#### 3.1.3. Standard Pull-Off Test for the Copper Coating

Although researchers typically deposit copper electrodes by vacuum deposition or attach the copper tape directly to the triboelectric layer to form a TENG, electrodes grown by chemical methods have unparalleled advantages. The main advantage of this technique is its ability to produce an ultra-thin adhesive layer (only 1–10 nm thick) (Figure A1), a polymer brush-based adhesive layer between the metal electrodes and the plastic substrate. Thus, there is no soft and uncontrollable adhesive material between the fragile metal electrode and the plastic substrate. We know that metals and polymer films are two incompatible materials; they have different surface energies and Young’s moduli. If the metal electrode film is directly evaporated on the flexible plastic substrate by vacuum deposition techniques, the electrode will extremely readily chip off and cannot pass the scotch tape test, which is approved by our previously reported PAMD-based Cu electrode [31]. Similarly, if we use the Cu tape as the electrode, it also cannot pass the scotch tape test because the copper tape also uses pressure-sensitive adhesive to attach to the triboelectric layer. 

Passing the adhesive tape test is the minimum requirement for the adhesion force between the metal electrode and the plastic substrate for our adhesive-tape-based TENG because the adhesive is also used as the contact interface. If the electrodes we manufacture are not strong enough, the entire device will fail during the continuous bonding and separation of the two triboelectric parts.

Fortunately, the electrode produced by this chemical method (i.e., the PAMD technique) can easily pass the adhesive tape test [31]. In this article, we have used a strong adhesive tape test to perform the pull-off test to further characterize the bond strength between the chemically produced metal electrode and the plastic substrate. A pull-off test can be used to assess the suitability of various coatings, paints, films and other thin materials [45]. If the coating has a weak adhesion force to the substrate, the coating will form cracks or delaminate from the substrate after the pull-off test. This method has been widely used to test the reliability of metal coatings, such as thin-film copper electrodes in printed circuit boards. As shown in Figure 3, the adhesion force between the metal electrode and the plastic substrate exceeded the upper limit of this test method, which further demonstrating its excellent adhesion properties and strong application potential. During the test, we prepared copper coatings on a 250-µm-thick PET film and divided the coating into 99 parts using a sharp tweezer. We then tried to determine the peel strength of the copper coating using adhesive tape (19 mm wide and peel strengths between 220 and 670 g/cm) according to European standard EN 16602-70-13, as shown in Figure 3. As required by the test procedure, the tape was first compressed onto the copper coating samples at a load of 22.5 MPa. Tension was then be applied to the tape to cause instantaneous separation between the scotch tape and the Cu coating at a constant pull-out speed (5 cm/min). The maximum tear force was found to be 0.7 MPa. The pressure and pull-off strength curve during the pull-off test is shown in Figure A2. After the pull-off test, the 99 separate areas of the copper coating remained attached to the plastic film; none of the coatings was removed by the adhesive tape, as shown in Figure 3c. The magnified image of the copper coating after the pull-off test shows that the coating remains constant, with no cracking or delamination, although some of the adhesive was torn off from the tape and adhered to the copper coating due to the great pressure and instantaneous peeling force, as shown in Figure 3d.

#### 3.1.4. Characterization of the Chemically Prepared Copper Layer

The chemically prepared copper coating on the PET film was found to be extremely resistant to damage. As shown in Figure 4, its resistance increased slightly and gradually approached a limit value, even when the slide rail repeatedly compressed the plastic film with the copper coating into a narrow space with a width of only ~2 mm. The surface roughness of the as-formed copper coating was characterized by scanning probe microscopy (SPM), and the root mean square roughness (Rq) of the as-formed electrodewas ~3 nm. The surface of the as-formed coating was found to be dense, smooth, and highly conductive. Using the four-probe method, we measured the surface resistivity of the copper coating to be less than 1 Ω/□.

### 3.2. Pressure-Sensitive Adhesive to Increase the Real Contact Area of the TENG

#### 3.2.1. The Pressure-Sensitive Adhesive and Real-Contact Area of the TENG

PDMS, the material mainly used to fabricate deformable triboelectric interface, is a polymeric organosilicon compound that has extremely low surface energy; as such, this material does not bind reliably to other materials. It is common for PDMS to become detached from substrates to which it is bonded during large deformations of the substrate. To make matter worse, PDMS is not strong and is easily shattered by strong impacts. Rubber (poly-cis-isoprene), rubber has many advantages and is likely to replace PDMS as a high performance, low-cost triboelectric interface. Rubber is a polymer synthesized from petroleum by-products. This durable and soft material inexpensive and widely available; it could be an ideal replacement for the PDMS typically used in TENGs.

Pressure-sensitive adhesive made by rubber shows good adhesion to PE films after hot-pressing treatment. After such treatments, the rubber rarely separates from the PE film. The adhesive used here also has excellent flexibility and conforms to the shape of the opposite surface of the PET, resulting in a large real contact area, as shown in Figure 5a.

The surface of most interfaces is rough; there are tiny bumps on the surface of ordinary objects, although the raised points are difficult to see with the naked eye or optical microscopes at low magnification, as shown in Figure A3. As a proof-of-concept, atomic force microscopy (AFM) was used to evaluate the surface morphology of the PET film; the results are shown in Figure 5b. Many raised dots appear on the surface of the PET, and the maximum size of these raised dots can exceed 200 nm in height. Without special polishing treatment, the surface of ordinary objects is rough.

Unfortunately, the presence of the raised points and particles on the surface of the triboelectric material prevents good contact between the two triboelectric materials. It is hypothesized that at least one of the friction materials must be soft in order to create a good fit with the second material through small deformations induced by external pressure. These deformations can lead to a significant increase in the real contact area at the interface. As shown schematically in Figure 5c, when the two materials are brought close together, only the raised points of the surface contact the polymer material on the opposite side. What’s more, the inevitable presence of dust and other contaminants in the environments in which TENGs are used also reduces their effective contact area during triboelectric activities. Such phenomena will reduce the power generation efficiency of most TENGs. Fortunately, we have developed a soft and durable adhesive that increases the effective contact area between the two triboelectric materials and increases the output signal by conforming to the shape of the intrinsic or introduced particles present in the triboelectric interface.

As a proof of concept, we carried out an experiment to compare the power generation of TENGs built with and without the presence of the pressure-sensitive adhesive in the contact area.

Our test involves manually contacting and separating the two triboelectric parts.The electrical signal was collected using a source meter (Keithley 2400 Sourcemeter). Figure 5e–f compares the short-circuit current (ISC) [45] of the as-built TENG device with and without the addition of the adhesive in the contact area. An almost 10-fold difference was observed in the maximum output current produced by the two TENG devices. The instability of the experimental signal may be due to the introduction of the biological signals when the finger touches the TENG electrodes to complete the experiment.

#### 3.2.2. The Output Performance of the As-Formed TENG

As shown in Figure 6a, the open circuit voltage (VOC) of the durable and flexible TENG can be between −3 V and 5 V with AC behavior due to the fast manual contact separation operation. Figure 6b shows the power density verification as the external load resistances change from 1 kΩ to 4 MΩ, and the maximum output power density is around 7.7 mW/m^2^. The instability of the experimental signal may be due to the introduction of the biological signals when the finger touches the TENG electrodes to complete the experiment.

#### 3.2.3. Special Features of the TENG with the Adhesive in the Contact Area

Compared to the PDMS often used in TENGs, the adhesive used here has higher adhesive strength to bond the two elements of the TENG. It is, therefore, relatively difficult to separate the two elements, especially in the case of wearable and flexible triboelectric films. When the tear speed is low (e.g., 8 mm/s), a unique output current pattern is generated, as shown in Figure 7. This output is slightly different from that of conventional TENGs [46]. However, the shape of the output generated in the as-formed TENG does not affect its power generation or applicability.

The existence of this increased adhesion between two triboelectric parts indicates that the two triboelectric materials are in good contact and that significant van der Waals interactions are formed between the surfaces, thus maximizing power generation. The formed TENG requires a slightly greater external force to separate the two triboelectric parts in the power generation process compared to PDMS-based devices, but the power that can be generated is also increased. This eliminates the need for an large external force to press the two friction plates together. The two triboelectric parts are easily bonded by the presence of a pressure-sensitive adhesive. Only a little more force is required to separate the two friction plates, but the increase in separation force is still small compared to forces generated by the movement of human limbs. This wear discomfort will be reduced with increased use or embedding of external contaminants, or with the invention and application of new low-adhesion pressure-sensitive adhesives. Increased adhesion will, therefore, not have a significant impact on wearer comfort.

### 3.3. Fabrication of Washable Garment-Integrated TENG (G-TENG) 

#### 3.3.1. Fabrication of Durable and Washable G-TENG

Washability is also a significant issue in G-TENG applications. The washing process is often accompanied by high torque, shear, and abrasion stresses that can damage or destroy the brittle structure of common electronic devices [45].

Here, the TENG device as designed with the use of the durable chemically deposited metal electrode and the durable triboelectric interface, has an advantage in terms of washability. As a proof of concept, we tested the washability of our proposed TENG using a washing machine (Summe, Hong Kong), working in normal mode (45 min with washing and spinning). The current output value was maintained at ~80% of the original value after the first cycles of machine washing and drying, as shown in Figure 8a. With repeated washing, it was found that the output current remained stable. It is hypothesized that the initial reduction in current output is due to the outer surface of the adhesive being contaminated with impurities from the water, or the adhesive undergoing slight surface modification during the water-washing process. However, after multiple washes, the signal output from the TENG gradually stabilizes. This indicates that the surface modification of the adhesive by the washing process has reached a certain equilibrium and is no longer changing. It also confirms that the back electrode, i.e., the chemically prepared metal electrode, is not affected by the washing process. This has also been confirmed by the work of our group in the past [36]. Figure 8b shows the as-formed flexible and durable G-TENG embroidered on a woven fabric. The triboelectric material covered with the back electrode is loosely sewn to the garment, ensuring that the triboelectric material faces upward. The copper electrode can be quickly welded to the copper-enamelled wire to form a conductive path. A paper-based object is placed under the triboelectric film to ensure good contact between the two triboelectric parts. This pad can be removed before each wash cycle and replaced after the wash.

#### 3.3.2. The Applicability of the G-TENGs in Health Monitoring

The TENG developed here can be used to power various electronic devices, such as health sensors and signal antennas, embedded in smart clothing, as shown in the diagram of potential applications (Figure 9). It can also function as a self-powered health sensor that can continuously record information about a patient’s physical activity and exercise over a long period of time. A buzzer and light-emitting diode can also be powered by the electrical energy generated by movement during the day to trigger an alarm in an emergency.

## 4. Discussion and Conclusions

Flexible G-TENGs harvest electrical energy from people’s body movements, but they lack durability. Although hard skeletal structures can be used to extend the life of the device, they are not compatible with thin and flexible clothing. However, clothing materials must be able to withstand high external forces, such as torques exceeding 8 kg·m^2^/s^2^ during machine washing, and pressures >100 KPa (e.g. an adult male weighing 75 kg can use his elbows to support half his body weight, with only ~36 cm^2^ of contact area between the elbow and the PDMS). We have pioneered the use of extremely durable materials in the manufacture of thin-film G-TENGs to extend the life of the device. In the manufacturing process, we first replaced the traditional thermally evaporated metal electrode with a novel durable electrode produced by chemical methods. The as-formed thin-film metal electrodes were electroless deposited onto an ultra-thin polymer-brush-based adhesive layer (1–10 nm thick) pre-embedded in plastic substrates. This paper focuses on the durability of the material itself and the high performance of the conductive electrodes, which remain robust without cracking or delaminating under a pressure of 22.5 MPa and a tear force of 0.7 MPa. In addition, the electrodes also have a smooth surface morphology and excellent electrical conductivity. We then used a low-cost but durable pressure-sensitive adhesive to create a reliable contact interface that can adapt to complex environments and increase the output current of the device. Due to the high deformability of the pressure sensitive adhesive, each friction of the TENG forms an almost complete fit at the triboelectric interface. This means that the real contact area for each contact is close to the maximum. The TENG produced here also exhibited high electrical stability under large mechanical deformations during the washing test.

At the same time, the process is cost-effective because the entire electrode manufacturing process takes place under ambient conditions. Extreme preparation conditions such as high temperature or high-energy curing environments are not required throughout the manufacturing process. The energy-saving nature of this technology can help the future industry of electrode preparation to adapt to the time of energy crisis.With the low production cost and energy consumption, this reported durable and flexible G-TENG could become a reliable and necessary energy harvesting unit for future smart clothing. 

## Figures and Tables

**Figure 1 sensors-23-02021-f001:**
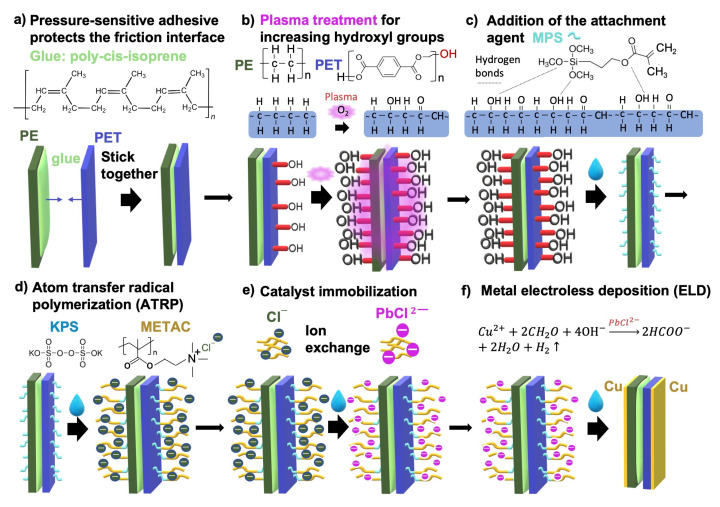
Schematic of the steps involved in simultaneous preparation of the dual electrodes on the flexible TENG. (**a**) First, a thin film of pressure-sensitive adhesive is used to protect the contact interface from contamination by the subsequent polymer-assisted metal deposition process. (**b**) Hydroxyl groups are then generated on the exposed PE and PET surfaces (by exposure to oxygen plasma); this process increases the wettability of the polymer surface. (**c**) Addition of the adhesion promoter, 3-(trimethoxysilyl)propyl methacrylate (MPS), which has a silane group that allows it to adhere to the smooth surface of the polymer layers and an acrylate group that enables subsequent graft polymerization. (**d**) Formation of the cationic brush by Atom Transfer Radical Polymerisation (ATRP) of METAC using the initiator potassium persulfate (KPS). (**e**) The anionic catalyst species can then be attached to the formed polymer brushes to initiate (**f**) electroless metal deposition of copper in the electroless bath.

**Figure 2 sensors-23-02021-f002:**
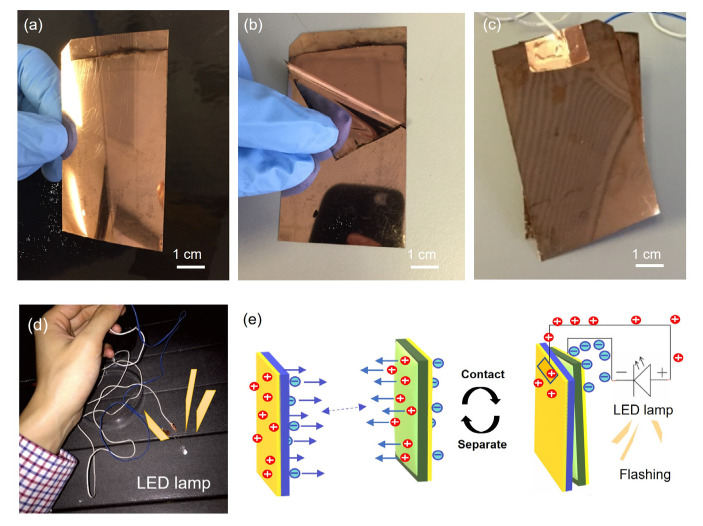
Photographs showing the two triboelectric layers originally bonded together (**a**) before separation, (**b**) during separation, and (**c**) after separation. The outer surface of the two triboelectric layers, i.e., the PET and PE layers, was coated with a uniform and highly adhesive thin copper film produced by a chemical process. (**d**) Photograph showing an illuminated LED lamp powered by a flexible and robust TENG. (**e**) Schematic of the mechanism behind the flexible TENG, which can be used to power an LED lamp.

**Figure 3 sensors-23-02021-f003:**
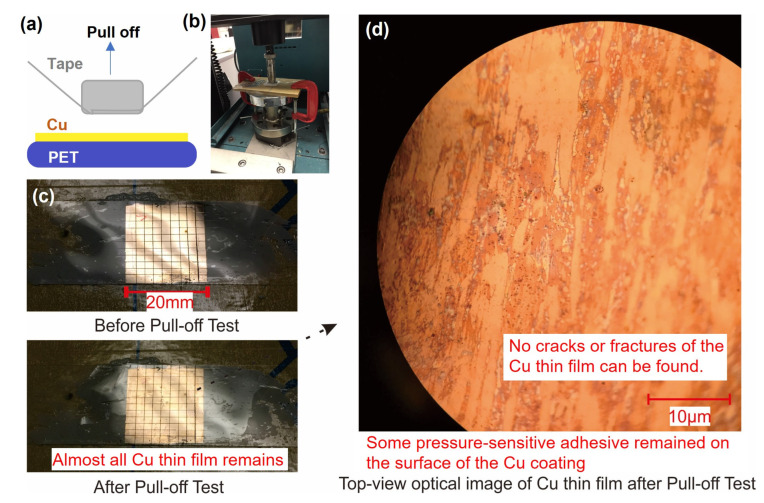
The peel test shows the extent of adhesion between the copper coating and the PET substrate. (**a**) Schematic showing how the peel test was performed. (**b**) Photograph of the setup used in the peel test. (**c**) Optical images showing the copper coating on the PET substrate before (**top**) and after (**bottom**) the peel test. (**d**) A microscopic image showing the surface of the copper coating. We used a ruler and sharp tweezers to make several parallel scratches on the surface of the copper coating to divide the original complete square Cu film into 99 parts. (**d**) A microscopic image showing the surface of the copper coating after the pull-off test.

**Figure 4 sensors-23-02021-f004:**
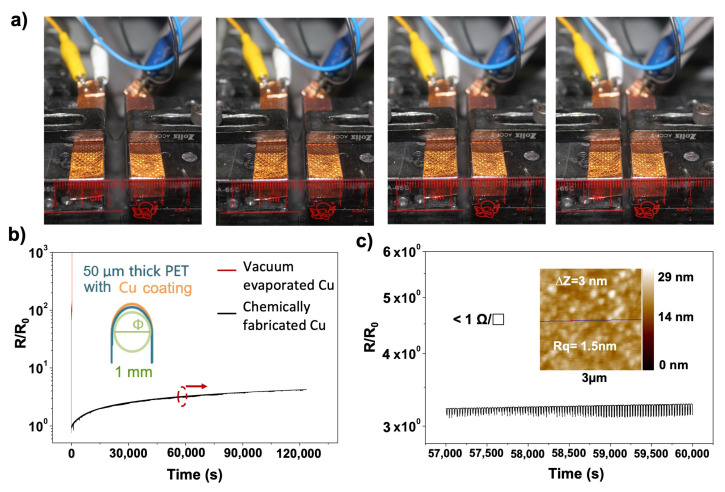
(**a**) Photograph of the chemically prepared copper coating during the endurance test. The four images, from left to right, describe the process of gradually bending the sample, i.e., the chemically prepared Cu thin-film electrodes on a triboelectric dielectric film, at a radius of 1 mm. (**b**) Resistance test of the copper coating on a plastic substrate subjected to repeated bending for approximately 36 h. The inset shows a cross-section of the copper coating during the bending test. (**c**) Enlarged view of the variation of the electrode resistance over a time interval of about 3000 s; the time interval considered in (**c**) is indicated by the red circle in (**b**). The inset in (**c**) shows a Scanning Probe Microscopy (SPM) image showing the surface morphology of the as-formed copper coating.

**Figure 5 sensors-23-02021-f005:**
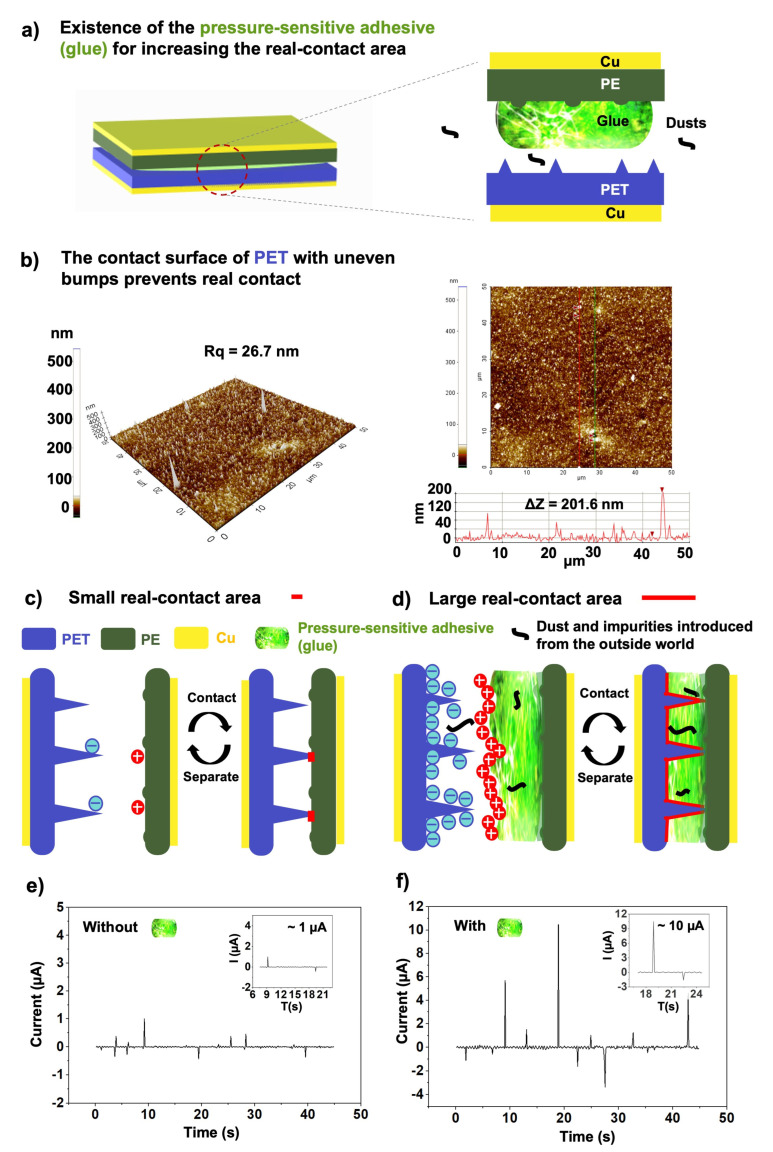
The pressure-sensitive adhesive used for increasing the real contact area of the TENG. (**a**) Schematic showing the TENG with a triboelectric interface formed by pressure-sensitive adhesive. (**b**) Atomic force microscopy (AFM) images and cross-sectional analysis showing the surface morphology of a normal PET film. (**c**,**d**) Schematic showing how the adhesive increases the real contact area when two normal polymer surfaces are in contact with a plastic substrate with a normal surface roughness. The output current of the TENG (**e**) with and (**f**) without the adhesive.

**Figure 6 sensors-23-02021-f006:**
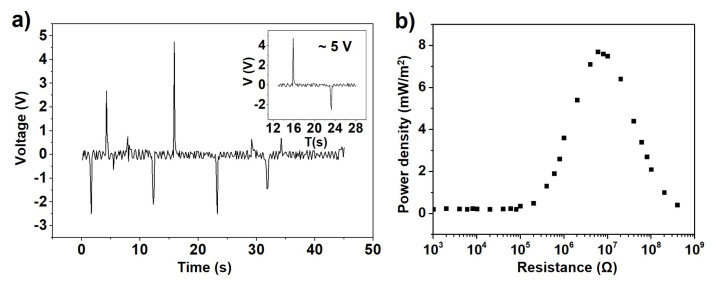
(**a**) The output voltage of the as-formed durable TENG device. (**b**) The relationship between power density and load resistance for the as-formed durable TENG.

**Figure 7 sensors-23-02021-f007:**
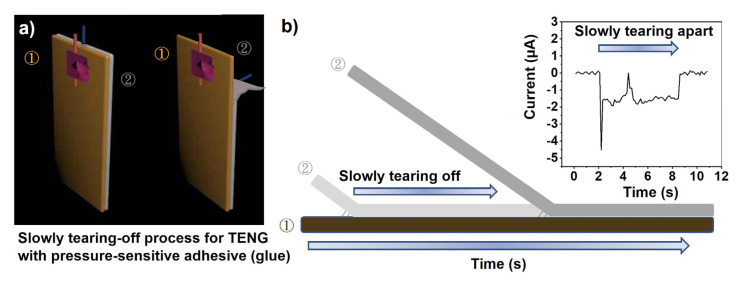
(**a**) Three-dimensional models and (**b**) a schematic showing the separation process of two triboelectric parts (i.e., ① and ②). The TENG has a triboelectric interface formed by the pressure sensitive adhesive. The inset of (**b**) shows the output current of the TENG during this process.

**Figure 8 sensors-23-02021-f008:**
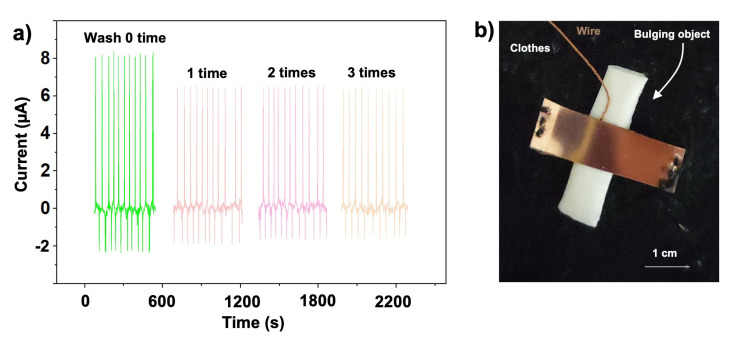
(**a**) The output current (**a**) shows a slight degradation in performance during garment washing. (**b**) The optical image of the as-formed durable and flexible TENG sewn into a garment.

**Figure 9 sensors-23-02021-f009:**
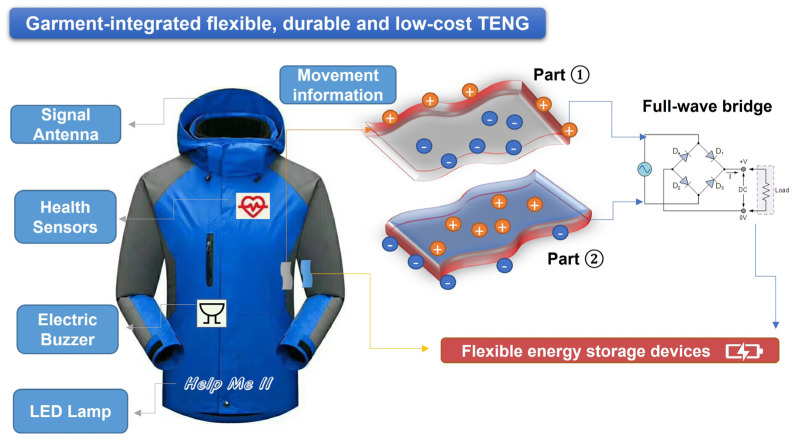
The potential applications of the flexible and robust G-TENG. To facilitate storage, a rectifier bridge can be added to the system to convert alternating current into direct current, and a flexible energy storage device can be used to collect the electrical energy harvested from everyday activities.

## Data Availability

The data presented in this study are available upon request from the corresponding author. The data are not publicly available due to the restriction of the equipment supplier.

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
