# Peer review of "Flexible and Robust Triboelectric Nanogenerators with Chemically Prepared Metal Electrodes and a Plastic Contact Interface Based on Low-Cost Pressure-Sensitive Adhesive"

_sensors, 2023, doi:10.3390/s23042021_

Round 1

Reviewer 1 Report

This work mainly reports a chemical preparation technique to fabricate metal electrodes and a pressure-sensitive adhesive-based TENG for mechanical energy harvesting. Compared with the previous work, it does have some progress. However, the polymer-assisted metal deposition is a commonly used method to prepare metal electrodes, and the figures are poorly prepared. The writing of the Introduction should be rewrote, which is too long and lack of logic. In addition, there are many errors in this manuscript and it is necessary to check the manuscript. So I would not suggest publishing it in this journal.

Reviewer 2 Report

In this work, the author developed a chemical preparation technique to fabricate durable and flexible metal electrodes via a polymer-assisted metal deposition technique, and then applied the highly adhesive electrode in the TENG applications. This manuscript can be improved after major revision, some points should be considered before publication as follows:

1.      In TENG manufacturing, researchers normally deposit copper electrode by vacuum deposition method or directly attach copper tape onto triboelectric layer to have a highly -adhesive electrode. And these methods could already provide a stable triboelectric layer-electrode interface, why bother proposing such a complex approach? What is the advantage of this technique

2.      Is the device designed based on contact-separation mode, how does charge transfer if the two triboelectric layer are bonded using pressure adhesive? Also, I doubt the results in Figure 5c-f, why fewer charges (blue dots)are generated on PET when no glue is applied to the two plastics (Figure 5c), and why more changes (red dots) are generated when there’s adhesive between PET and PE (Figure 5d).

3.      In Figure 5c-d, the charges should be positive and negative charges instead of all negative charges.

4.      Could you provide peel-strength curve when investigating the adhesion of the flexible metal electrodes via a polymer-assisted metal deposition technique?

5.      Could the high adhesive of metal electrode be applied to other materials?

6.      The durability test of the TENG device should be provided.

7.      Legends of the curve should be added to Figure 4b.

8.      The chemical reaction could be incorporated in Figure 1 to provide more direct information for readers to understand.

9.      Why the current and voltage put of the device is not stable (Figure 5c 5f 6a)

10.   Several references discussing TENG with high adhesive should be added: ACS Nano 2018, 12, 2818; Matter 2021, 4, 1962. ect.

Round 2

Reviewer 1 Report

The quality of this manuscript including the English language and figures has been improved significantly, and I think it can be accepted now.

Reviewer 2 Report

The authors have provided detailed point-to-point regarding my questions and concerns. I think it could be accepted by Sensors.